# Idiographic Personality Gaussian Process for Psychological Assessment

**Yehu Chen, Muchen Xi, Jacob Montgomery**
**Joshua Jackson, Roman Garnett**
Washington University in St Louis
chenyehu,m.xi,j.jackson,jacob.montgomery,garnett@wustl.edu

## Abstract

We develop a novel measurement framework based on a Gaussian process coregionalization model to address a long-lasting debate in psychometrics: whether psychological features like personality share a common structure across the population, vary uniquely for individuals, or some combination. We propose the idiographic personality Gaussian process (IPGP) framework, an intermediate model that accommodates both shared trait structure across a population and "idiographic" deviations for individuals. IPGP leverages the Gaussian process coregionalization model to handle the grouped nature of battery responses, but adjusted to non-Gaussian ordinal data. We further exploit stochastic variational inference for efficient latent factor estimation required for idiographic modeling at scale. Using synthetic and real data, we show that IPGP improves both prediction of actual responses and estimation of individualized factor structures relative to existing benchmarks. In a third study, we show that IPGP also identifies unique clusters of personality taxonomies in real-world data, displaying great potential to advance individualized approaches to psychological diagnosis and treatment.

## 1   Introduction

Building models for the evaluation of latent traits from observed responses is crucial to understand long-term behaviors through repeated quantitative assessments. These are used, for example, to study emotional stability after medical treatment or the development of academic ability during secondary education [1–3]. However, existing frameworks face several interrelated limitations. First, there are strong reasons to believe that standard taxonomies may over-generalize, failing to distinguish between related psychological phenomena that often differ in etiology, symptoms, and biological processes between individuals [e.g., 1, 4], and this may lead to inaccuracy when making predictions [5]. A related issue is that measurement models are rarely individualized, instead assuming that (1) the correlation between latent traits of interest and survey responses are invariant across individuals and (2) the relationship between the latent traits themselves are the same for everyone. Lastly, current models are almost always developed for cross-sectional data that are collected only once from each respondent, which overlooks any potential dynamics of psychological processes.

To address these limitations, previous research has adopted three different approaches, each inadequate in its own way. First, recent work has proposed an *idiographic* approach that builds a completely distinct taxonomy for each individual [6–8]. However, complete personalization can sacrifice generalizability and interpretability for clinicians, as any possible population commonality is ignored. A second line of research focuses on building dynamic psychometric models of time-series data via some variant of item response theory [9, 10, 3], longitudinal structural equation modeling [11–13], vectorized autoregression [14, 15] and/or Gaussian process (GP) latent trajectories [16–18]. However, all these models adopt the *nomothetic* approach, assuming that the responses of all individuals

share an identical latent structure. Finally, there is a smaller body of work that adopts intermediate approaches to create individualization while maintaining group commonality [e.g., 19]. However, prior research models quantitative responses directly, ignoring the latent structures that are often the actual focus of domain researchers.

In this work, we propose an idiographic personality Gaussian process (IPGP) framework for assessing dynamic psychological taxonomies from time-series survey data. This framework combines nomothetic and idiographic approaches by employing a common structure to explain typical circumstances, while allowing individual structures to accommodate deviations into distinct forms. We leverage the Gaussian process coregionalization model based on multi-task kernels to conceptualize responses of grouped survey batteries, adjusted to non-Gaussian ordinal data, and utilize IPGP for hypothesis testing of domain theories. Methodologically, our approach combines Gaussian process latent variable models (GPLVM) [20], Gaussian process dynamic systems (GPDM) [17, 18] and GP ordinal regression for Likert-type survey data [21, 22]. Despite involving latent variables and GPs, IPGP differs from GPLVM in two ways: (1) it optimizes the factor loading matrix while marginalizing the latent variables, and (2) it accommodates categorical data using a non-Gaussian ordered probit likelihood. Computationally, our framework exploits stochastic variational inference for latent factor estimation, contrasting with other GP measurement models relying on Gibbs sampling that may not scale efficiently to intensive longitudinal setups [18, 23].

To our knowledge, our work presents the first multi-task Gaussian process latent variable model for dynamic idiographic assessment. While multi-task GPs have found recent applications in areas such as causal inference [24–26], environmental science [27–29], and biomedical research [30], their potential remains largely unexplored in psychology. Current psychometric models typically focus on cross-sectional settings without dynamics [6, 31, 8] or single task settings that ignore inter-battery correlations [32, 33]. We conducted an extensive simulation study comparing IPGP against benchmark methods and analyzed an existing cross-sectional personality dataset. Our results demonstrate that IPGP simultaneously enhances the estimation of idiographic taxonomies and improves the prediction of responses. Additionally, we collected a novel IRB-approved longitudinal dataset. When applied to this data, IPGP not only shows superior performance in response prediction, but also suggests unique personality taxonomies. These findings highlight IPGP's significant potential for advancing individualized approaches to psychological diagnosis and treatment.

## 2  Background

We start by laying out the ordinal factor model for building standard taxonomy from survey data [34, 35]. We then briefly discuss several existing idiographic longitudinal models in psychological assessment and review the Gaussian process model.

**Ordinal factor analysis.**  Consider the scenario where some set of units, $i \in \{1, \dots, N\}$, repeatedly answer the same set of $j \in \{i, \dots, J\}$ survey items over $t \in \{1, \dots, T\}$ periods with ordinal observations $y_{ijt} \in \{1, \dots, C\}$ up to $C$ levels. For example, the responses could be Likert-typed, ranging from "strongly disagree" to "strongly agree". The latent factor model posits that the $j$th underlying latent variable $f_j^{(i)}(t)$ for unit $i$ at time $t$ can be recovered as $\mathbf{w}_j^T \mathbf{x}_i(t)$, where $\mathbf{x}_i(t) \in \mathbf{R}^D$ are unit-level latent factors and $\mathbf{w}_j \in \mathbf{R}^D$ are factor loadings. The $f_j^{(i)}(t)$s are then mapped to ordinal responses via an ordered probit model: $p\big(y_{ijt} = c \mid f_j^{(i)}(t) = f\big) = \Phi(b_c - f) - \Phi(b_{c-1} - f)$ with threshold parameters $b_0 < \cdots < b_C$. Usually $b_0$ and $b_C$ are fixed to $-\infty$ and $+\infty$ so that the resulting categorical probability vector sums to 1, while $b_1, \dots, b_{C-1}$ are allowed to move freely. Stacking $\mathbf{x}_i(t)$s, $\mathbf{w}_j$s and $y_{ijt}$'s into matrices $\mathbf{x}$, $\mathbf{w}$ and tensor $\mathbf{y}$, the joint likelihood can be written as $\mathcal{L}(\mathbf{y} \mid \mathbf{x}, \mathbf{w}) = \prod_i \prod_j \prod_t p\big(y_{ijt} \mid \mathbf{x}_i(t), \mathbf{w}_j\big)$. Identification is guaranteed with standard orthogonality and normalization constraints [36]. This factor model is also known as an item response model [37, 38], and can be estimated via maximum likelihood, weighted least squares, or an EM algorithm [39–41].

**Idiographic longitudinal assessment.**  In psychological assessment, the idiographic approach emphasizes *intrapersonal* variation by requiring distinct loadings, while the nomothetic approach identifies general *interpersonal* variation assuming shared factor loadings [42]. In terms of data collection, the idiographic approach usually surveys each individual multiple times ($N = 1$ and large

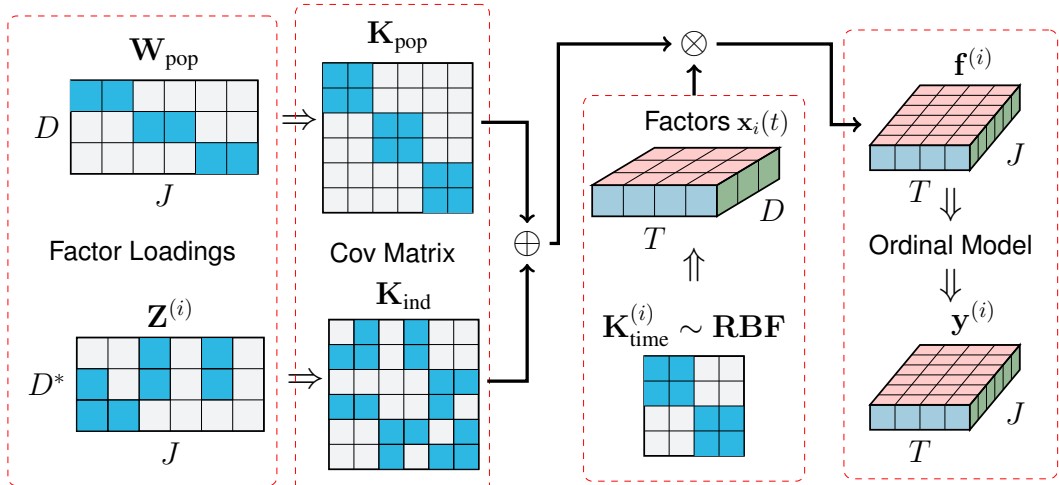

Figure 1: Proposed IPGP model for inferring latent factors and factor loadings from dynamic ordinal data. Input ordinal observations across indicators are modeled as ordinal transformations of latent dynamic Gaussian processes with individualized RBF kernels and loading matrices.

$T$) for learning personalized taxonomy rather than many individuals at a single shot (large $N$ and $T = 1$). To extract individualized dynamics from time-series data, recent psychometric models have utilized longitudinal structural equations by explicitly specifying any intrapersonal and temporal dynamics. However, these typically require strong model-based assumptions from domain-theory about this dynamic process, and may be sensitive to model misspecification [11, 13]. Meanwhile, hierarchical vector autoregression models may automatically learn individual trajectories over time, but are designed to model observed responses directly rather than latent traits of interest to domain scholars [14, 15].

**Gaussian process.** A Gaussian process (GP) can be used to define a distribution over $f$ such that the values of $f$ at arbitrary finite subset of $\mathcal{X}$ have a joint multivariate Gaussian [43]. A $\mathcal{GP}(\mu, K)$ is specified with a mean function $\mu\colon \mathcal{X} \to \mathbf{R}$ and a positive-definite kernel function $K\colon \mathcal{X} \times \mathcal{X} \to \mathbf{R}$; evaluating these functions pointwise determines the mean and covariance of these finite-dimensional distributions. The most common kernel is the squared exponential (RBF) kernel $K(\mathbf{x_1}, \mathbf{x_2}) = \exp\left(-\frac{1}{2}\mathbf{x_1}^T \mathbf{P} \mathbf{x_2}\right)$ with precision matrix $\mathbf{P} = \mathbf{diag}(1/\ell_1^2, \ldots, 1/\ell_d^2)$ and $d = \mathbf{card}(\mathcal{X})$. The posterior of a GP can be derived analytically for a Gaussian likelihood but must be approximated in modeling latent variables with categorical indicators. We discuss the variational approximation method we use for inference in Sec. (3).

## 3 Methodology

We propose an idiographic personality Gaussian process (IPGP) framework for assessing individualized dynamic psychological taxonomies from time-series survey data. Instead of joint estimation of latent factors and their loadings that cannot guarantee rotational and scaling invariance, we marginalize out the latent variables and focus on learning taxonomies of loadings. The overall architecture of IPGP is illustrated in Figure (1), where the ordinal input observations across the indicators are modeled as ordinal transformations of latent dynamic GP with individualized RBF kernels and loading matrices.

### 3.1 Multi-task learning

Typically in psychological assessment, survey questions are meticulously grouped such that each group gauges a particular latent trait (e.g. dimension of personality). Hence, we conceptualize the assessment of psychological traits as a multi-task learning problem, where each question represents a distinct task but can be correlated with other tasks. A multi-task GP is an extension of the single-task GP but for vector-valued functions [31]. To motivate the multi-task framework, first consider the

two-task scenario with two $T \times 1$ vector $\mathbf{f_1}^{(i)}$ and $\mathbf{f_2}^{(i)}$ denoting the latent temporal processes of unit $i$ for question $j = 1, 2$. To fix the scale of latent factors, a time-level Gaussian process prior is placed on $\mathbf{x}_i(t) \sim \mathcal{GP}(\mathbf{0}, \mathbf{K}_{\text{time}}^{(i)})$. By exploiting affine property of Gaussians, the induced joint distribution of vectorized $[\mathbf{f_1}^{(i)}, \mathbf{f_2}^{(i)}]^T$ can be written as:

$$\begin{bmatrix} \mathbf{f_1}^{(i)} \\ \mathbf{f_2}^{(i)} \end{bmatrix} \sim \mathcal{GP}\Big( \begin{bmatrix} \mathbf{0} \\ \mathbf{0} \end{bmatrix}, \begin{bmatrix} \mathbf{w}_1^T \mathbf{w}_1 \mathbf{K}_{\text{time}}^{(i)} & \mathbf{w}_1^T \mathbf{w}_2 \mathbf{K}_{\text{time}}^{(i)} \\ \mathbf{w}_2^T \mathbf{w}_1 \mathbf{K}_{\text{time}}^{(i)} & \mathbf{w}_2^T \mathbf{w}_2 \mathbf{K}_{\text{time}}^{(i)} \end{bmatrix} \Big) \tag{1}$$

whose covariance of shape $2T \times 2T$ contains four block matrices $\mathbf{K}_{\text{time}}^{(i)}$ scaled by different $\mathbf{w}_j^T \mathbf{w}_{j'}$ ($j, j' \in \{1, 2\}$). Specifically, $\mathbf{w}_1^T \mathbf{w}_2$ controls the inter-task covariance between these two tasks and $\mathbf{w}_j^T \mathbf{w}_j$s ($j \in \{1, 2\}$) control their intra-task variance. This multi-task structure is also known as the linear model of coregionalization (LMC) [44], which models output functions as linear combinations of several independent latent processes. In our case, each dimension in $\mathbf{x}_i(t)$ represents one latent process, which jointly defines the functions as $\mathbf{f_j}^{(i)} = \mathbf{w}_j^T \mathbf{x}_i(t)$. To extend this, let $\mathbf{f}^{(i)} = [\mathbf{f_1}^{(i)}, \ldots, \mathbf{f_J}^{(i)}]^T$ represents the flattened $JT \times 1$ vector consisting of all $J$ tasks. We write $\mathbf{f}^{(i)}$ in a formal multi-task GP notation using Kronecker product $\otimes$:

$$\mathbf{f}^{(i)} \sim \mathcal{GP}\big(\mathbf{0}, \mathbf{K}_{\text{task}}^{(i)} \otimes \mathbf{K}_{\text{time}}^{(i)}\big) \tag{2}$$

$$\mathbf{K}_{\text{task}}^{(i)} = \mathbf{W}_{\text{pop}}^T \mathbf{W}_{\text{pop}} + \mathbf{Z}^{(i)T} \mathbf{Z}^{(i)} \tag{3}$$

where $\mathbf{K}_{\text{task}}^{(i)}$ denotes the unit-specific task covariance matrix, consisting of the self inner products of $D \times J$ shared interpersonal loading $\mathbf{W}_{\text{pop}}$ and $D^* \times J$ unit-specific low-rank $\mathbf{Z}^{(i)}$ ($D^* < D$) for intrapersonal deviations that serves to be the additional idiographic component, independent of $\mathbf{W}_{\text{pop}}$. In our experiments, we found degraded performance for $D^* = 2$ but extra computational costs so we focused on $D^* = 1$. The Kronecker product $\otimes$ then multiplies each entry in the $J \times J$ task covariance with $\mathbf{K}_{\text{time}}^{(i)}$, and returns the stacked $JT \times JT$ covariance for $\mathbf{f}^{(i)}$. Through the use of time kernel $\mathbf{K}_{\text{time}}^{(i)}$, properties of the latent trait trends such as periodicity or autocorrelation could be incorporated. Here we use the common RBF kernel $\mathbf{K}_{\text{time}}^{(i)}(t, t') = \exp\big(-\frac{1}{2}(t - t')^2 / \ell_i^2\big)$ to account for dynamic changes in the latent attributes, whose bandwidth is determined by the unit-specific length scale $\ell_i$, but any other kernel can substitute RBF as practitioners see fit. Finally, the latent variables $\mathbf{f}^{(\mathbf{i})}$s are further projected to response space by the ordered probit model.

## 3.2 Variational inference

Due to the non-Gaussian ordinal likelihood, we adopt the variational inference technique (VI) with inducing points introduced in [33]. Dropping superscript for demonstration, VI utilizes a variational distribution $q(\mathbf{u}) = \mathcal{N}(\mu_{\mathbf{u}}, \mathbf{\Sigma}_{\mathbf{u}})$ on inducing variables $\mathbf{u}$ to approximate $p(\mathbf{f} \mid \mathbf{y})$ using the conditional $p(\mathbf{f} \mid \mathbf{u})$. Hence, the conditional log-likelihood $\log p(\mathbf{y} \mid \mathbf{u})$ can be lower bounded by the expected log-likelihood w.r.t. $p(\mathbf{f} \mid \mathbf{u})$, after exploiting the non-negativity of Kullback–Leibler (KL) divergence between $p(\mathbf{f} \mid \mathbf{u})$ and $p(\mathbf{f} \mid \mathbf{y})$:

$$\log p(\mathbf{y} \mid \mathbf{u}) \geq \mathbb{E}_{p(\mathbf{f}|\mathbf{u})} \log p(\mathbf{y} \mid \mathbf{f}) \tag{4}$$

Furthermore, a lower bound on model evidence (ELBO) can be obtained by combining Eq. (4) and an inequality derived by another KL divergence $\text{KL}[q(\mathbf{u}) \parallel p(\mathbf{u} \mid \mathbf{y})] \geq 0$ (see Appendix B for details):

$$\log p(\mathbf{y}) \geq \mathbb{E}_{q(\mathbf{u})}\big[\log p(\mathbf{y} \mid \mathbf{u})\big] - \text{KL}[q(\mathbf{u}) \parallel p(\mathbf{u})] \tag{5}$$

$$\geq \mathbb{E}_{q(\mathbf{f})}\big[\log p(\mathbf{y} \mid \mathbf{f})\big] - \text{KL}[q(\mathbf{u}) \parallel p(\mathbf{u})] \tag{6}$$

where the KL divergence $\text{KL}[q(\mathbf{u}) \parallel p(\mathbf{u})]$ between the variational $q(\mathbf{u})$ and prior $p(\mathbf{u})$ can be computed in closed form as both distributions are Gaussians. The expectation of log-likelihood $\log p(\mathbf{y} \mid \mathbf{f})$ under the marginal distribution $q(\mathbf{f}) = \int p(\mathbf{f} \mid \mathbf{u}) q(\mathbf{u}) d\mathbf{u}$ is intractable but can be numerically approximated using Gauss–Hermite quadrature method. The variational parameters $\mu_{\mathbf{u}}$ and $\mathbf{\Sigma}_{\mathbf{u}}$, individualized loadings $\mathbf{w}_i$ and $\text{diag}(\mathbf{v})$ as well as likelihood parameters $\{b_c\}$s are then optimized to maximize this lower bound. Finally, the predictive likelihood of new $p(\mathbf{y}^*) = \int p(\mathbf{y}^* \mid \mathbf{f}^*) p(\mathbf{f}^* \mid \mathbf{u}) q^*(\mathbf{u}) d\mathbf{u}$ is obtained by marginalizing out the optimized $q^*(\mathbf{u})$. Throughout our experiments, we adopt stochastic inference for computational scalability.

Table 1: Comparison of averaged accuracy, log-likelihood and correlation matrix distance between IPGP, baselines, and ablated models in the simulation study. The full IPGP model (indicated in bold) significantly outperforms all ablated and baseline methods. Results from ablations imply that IPGP succeeds in predicting the correct labels due to its idiographic components and proper likelihood, and a well-informed population kernel is crucial in recovering the factor loadings. "—" indicates baseline software that cannot handle missing values.

| | TRAIN ACC ↑ | TRAIN LL ↑ | TEST ACC ↑ | TEST LL ↑ | CMD ↓ |
|---|---|---|---|---|---|
| GRM | $0.261 \pm 0.005$ | $-3.556 \pm 0.092$ | $0.261 \pm 0.006$ | $-3.578 \pm 0.098$ | $0.657 \pm 0.021$ |
| GPCM | $0.562 \pm 0.017$ | $-2.067 \pm 0.182$ | $0.495 \pm 0.012$ | $-2.409 \pm 0.143$ | $0.545 \pm 0.016$ |
| SRM | $0.286 \pm 0.006$ | $-7.408 \pm 0.063$ | $0.289 \pm 0.008$ | $-7.341 \pm 0.084$ | $0.300 \pm 0.024$ |
| GPDM | $0.687 \pm 0.010$ | $-4.358 \pm 0.028$ | $0.667 \pm 0.010$ | $-4.377 \pm 0.029$ | $0.262 \pm 0.016$ |
| LSM | $0.539 \pm 0.021$ | $-0.961 \pm 0.015$ | — | — | $0.256 \pm 0.011$ |
| TVAR | $0.554 \pm 0.018$ | $-1.168 \pm 0.014$ | — | — | $0.987 \pm 0.013$ |
| IPGP-NOM | $0.807 \pm 0.007$ | $-0.535 \pm 0.015$ | $0.790 \pm 0.008$ | $-0.555 \pm 0.017$ | $0.257 \pm 0.009$ |
| IPGP-IND | $0.932 \pm 0.003$ | $-0.243 \pm 0.008$ | $0.916 \pm 0.004$ | $-0.267 \pm 0.009$ | $0.530 \pm 0.005$ |
| IPGP-LOW | $0.897 \pm 0.004$ | $-0.313 \pm 0.010$ | $0.884 \pm 0.005$ | $-0.334 \pm 0.011$ | $0.397 \pm 0.007$ |
| IPGP-NP | $0.898 \pm 0.003$ | $-0.318 \pm 0.009$ | $0.883 \pm 0.005$ | $-0.342 \pm 0.011$ | $0.467 \pm 0.010$ |
| **IPGP** | $\mathbf{0.957 \pm 0.002}$ | $\mathbf{-0.159 \pm 0.005}$ | $\mathbf{0.942 \pm 0.002}$ | $\mathbf{-0.184 \pm 0.006}$ | $\mathbf{0.128 \pm 0.006}$ |

## 3.3 Theory testing

Our IPGP framework also naturally facilitates downstream tasks such as domain theory testing between models with and without shared or idiographic components. We adopt posterior odds ratio test, using posterior $p(\mathcal{M}_i \mid \mathbf{y}) = \frac{p(\mathbf{y}|\mathcal{M}_i)p(\mathcal{M}_i)}{\sum_i p(\mathbf{y}|\mathcal{M}_i)p(\mathcal{M}_i)}$ over a pool of models $\{\mathcal{M}_i\}$ conditioning on observations $\mathbf{y}$ with prior weights $p(\mathcal{M}_i)$, as the hypothesis test to determine whether the latent structures for each individual are indeed distinct or are simply explainable by interpersonal commonality. Specifically, we refer the multi-task model in Eq. (3) as the *idiographic* model, and compare it with an *nomothetic* model without unit-specific components: $\mathbf{K}_{\text{task}}^{\text{pop}} = \mathbf{W}_{\text{pop}} \mathbf{W}_{\text{pop}}^T$.

Note that compared to this baseline nomothetic model, our proposed idiographic model in Eq. (3) introduces additional unit-level $JN$ loading parameters that enlarge the hyperparameter optimization space. Hence, we propose to first learn the interpersonal loading matrix $\mathbf{W}_{\text{pop}}$ using the standard cross-sectional data from a nomothetic model that focuses on learning of population taxonomy, and then use the estimated $\mathbf{W}_{\text{pop}}$ as informative prior in the full model. We will show empirically in Sec. (4) that with this stronger prior IPGP achieves a more precise estimation of individual taxonomies.

## 4 Experiments

We now evaluate IPGP in learning idiographic latent taxonomies and predicting actual responses against baseline methods from both psychometrics and Gaussian process literature in three experiments: a simulation study, a re-analysis of a large cross-sectional dataset, and a pilot study of repeated measures of the Big Five [45] personality traits.

### 4.1 Simulation and ablation

**Setup.** Our simulation considers longitudinal data of $N = 10$ units over $T = 30$ periods. We assume that the latent traits of each unit $i$ have dimension $D = 5$, and each latent vector is generated independently from a GP $\mathbf{x}_i^{(d)}(t) \sim \mathcal{GP}(\mathbf{0}, \mathbf{K}_{\text{time}}^{(i)})$ with a unit-specific length scale uniformly randomly picked from $\ell_{\text{time}}^{(i)} \in [10, 20, 30]$. We split $J = 20$ items into $D$ subsets of size $J/D = 4$, such that each subset dominates one dimensional in the latent traits. Specifically, we set high value of 3 in the population factor loading matrix $\mathbf{W}_{\text{pop}}$ for entries corresponding to the $k$th subset for dimension $k$, and low values drawn from Unif$[-1, 1]$ otherwise. We also set each unit-specific loading $\mathbf{w}_i$ from Unif$[-1, 1]$. To introduce sparsity and reverse coding, we randomly set half of the loadings to zero and invert the signs of the remaining half. Finally, we generate the $y_{ijt}$s according to the ordered probit model with $C = 5$ levels, and apply $80\%/20\%$ splitting for training and testing.

Table 2: Model comparison where the model rank varies from 2, 5 to 8 while the true rank is 5. The best models are indicated in bold, and models that are not significantly worse than the best model are indicated in italics.

| RANK | TRAIN ACC ↑ | TRAIN LL ↑ | TEST ACC ↑ | TEST LL ↑ | CMD ↓ |
|---|---|---|---|---|---|
| 2 (LOWER) | $0.897 \pm 0.004$ | $-0.313 \pm 0.010$ | $0.884 \pm 0.005$ | $-0.334 \pm 0.011$ | $0.397 \pm 0.007$ |
| 5 (TRUE) | $\mathbf{0.957 \pm 0.002}$ | $-0.159 \pm 0.005$ | $0.942 \pm 0.002$ | $-0.184 \pm 0.006$ | $0.128 \pm 0.006$ |
| 8 (HIGHER) | $\mathbf{0.957 \pm 0.002}$ | $\mathbf{-0.161 \pm 0.004}$ | $\mathbf{0.945 \pm 0.002}$ | $\mathbf{-0.183 \pm 0.005}$ | $\mathbf{0.124 \pm 0.006}$ |

**Metrics and baselines.** We consider two sets of metrics for evaluation: (1) the in-sample and out-of-sample predictive accuracy (ACC) and log-likelihood (LL) of the actual responses, (2) the correlation matrix distance (CMD) between the estimated factor loading matrix and the true ones, which is defined for two covariance matrices $\mathbf{R}_1, \mathbf{R}_2$ as $\mathrm{d}(\mathbf{R}_1, \mathbf{R}_2) = 1 - \frac{\mathrm{tr}(\mathbf{R}_1 \mathbf{R}_2)}{\|\mathbf{R}_1\|_f \|\mathbf{R}_2\|_f}$ [46] where $f$ is the Frobenius norm. Note that CMD becomes zero if $\mathbf{R}_1, \mathbf{R}_2$ are equal up to a scaling factor, and one if they are orthogonal after flattening. We compare IPGP to (1) various latent variable models for ordinal responses, including the graded response model (GRM) [37], the generalized partial credit model (GPCM) [47] and the sequential response model (SRM) [48], (2) Gaussian process dynamic model (GPDM) [17, 18] where the continuous predictions are rounded to the nearest ordinal level, (3) latent structural model (LSM) [13, 49] with trait-dependent latent variables and (4) time-varying vector autoregression (TVAR) with regularized kernel smoothing [15]. We also compare IPGP with several ablated models: (1) IPGP-NOM without the idiographic kernel, (2) IPGP-IND without the population kernel, (3) IPGP-LOW with lower-rank factors of 2 than actual rank of 5 in the synthetic setup and (4) IPGP-NP where the population kernel is learned from scratch rather than fixed to the informative prior. Note that $\mathbf{W}_{\mathrm{pop}}$ in the full IPGP model is fixed as learned from IPGP-NOM.

**Results.** We use 100 inducing points and the ADAM optimizer with learning rate 0.05 to optimize ELBO for 10 epochs with batch size of 256. We repeat our simulation with 25 different random seeds using 300 cores on Intel Xeon 2680 CPUs. Table 1 shows the comparison of the average predictive accuracy, log-likelihood, and correlation matrix distance between IPGP and baselines and ablated models. Our IPGP model (indicated in bold) significantly outperforms all ablated models and baseline methods in estimated correlation matrix, predictive accuracy, and log-likelihood of both training and testing sets in paired-$t$ tests. We found that IPGP is able to predict the correct labels due to its idiographic components and proper likelihood, since IPGP-NOM and IPGP-GL are two of the worst ablations for all prediction metrics. In addition, IPGP-IND and IPGP-NP have the worst correlation matrix estimation, implying that a well-informed population kernel is crucial in recovering the underlying factor structures.

**Robustness of IPGP.** To assess IPGP's robustness to rank misspecification, we conducted additional exploratory factor analysis using our simulated data. We tested model performance with ranks of 2, 5, and 8, where 5 represents the true rank of the data. As shown in Table 2, both the true-rank (5) and higher-rank (8) models significantly outperform the low-rank (2) model. However, increasing the rank beyond the true rank of 5 yields no additional performance benefits: the high-rank model is not significantly better than the true-rank model in a paired t-test. These results demonstrate two key points: first, low-rank approximations inherently lack the capacity to fully capture the underlying structure, and second, increasing the rank beyond the true rank provides no additional benefit. This underscores the importance of careful exploratory analysis in practical applications to determine the appropriate rank.

## 4.2 Cross-sectional factor analysis

We next validate the popular Big Five personality theory using standard cross-sectional data via a factor analysis, where a range of factors are tested and then compared according to model evidence. This serves to show that the model works appropriately to detect known latent traits even in non-dynamic settings, and to validate the informative prior for the $\mathbf{W}_{\mathrm{pop}}$ matrix in our next experiment. We utilize an existing dataset called life outcomes of personality replication (LOOPR) [50], which is collected from 5,347 unique participants on the Big Five Inventory [51] consisting of 60 questions. Our validation considers a range of latent trait dimension counts from $D = 1, \ldots, 5$. For each dimension count, we first apply principal component analysis (PCA) directly on the correlation matrix

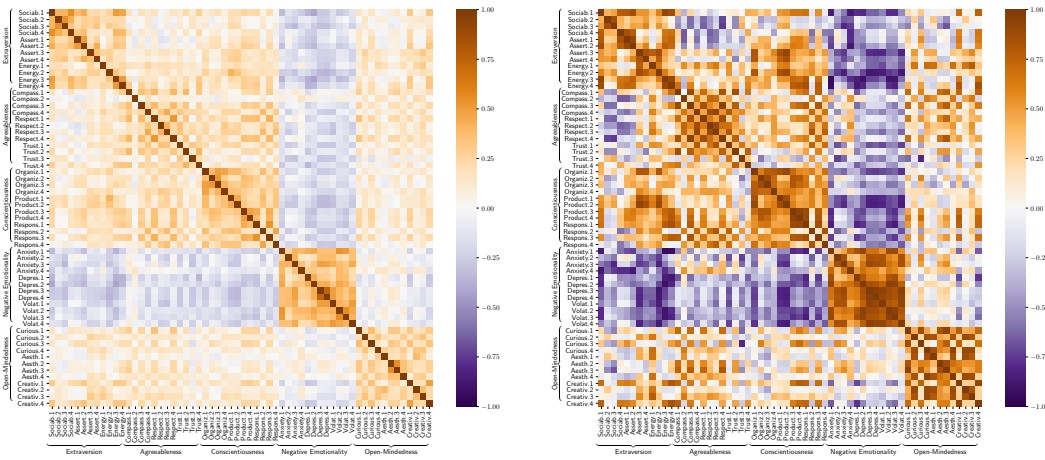

Figure 2: Illustration of raw correlation matrix (left) and our estimated Big Five loading matrix (right). Both correlation matrices display a *block* pattern, where estimated interpersonal variation show strong correlation between questions within the same factor of the Big Five personalities and weak correlation across different factors. Besides, questions corresponding negative emotionality show minor negative correlation with those corresponding to extraversion and conscientiousness, suggesting trait-by-trait interaction effects.

Table 3: In-sample accuracy and averaged log lik of our method and baselines for various ranks $D$ in LOOPR. Best model for each $D$ is indicated in bold and the best model across different $D$s is further indicated in italic.

| | ACC ↑ | | | | | LL / N ↑ | | | | |
| MODEL | $D = 1$ | $D = 2$ | $D = 3$ | $D = 4$ | $D = 5$ | $D = 1$ | $D = 2$ | $D = 3$ | $D = 4$ | $D = 5$ |
|---|---|---|---|---|---|---|---|---|---|---|
| PCA | 0.106 | 0.099 | 0.123 | 0.217 | 0.192 | $-1.957$ | $-1.990$ | $-2.009$ | $-2.036$ | $-2.051$ |
| GRM | 0.238 | 0.107 | 0.178 | 0.113 | 0.146 | $-1.838$ | $-1.832$ | $-1.814$ | $-1.838$ | $-1.841$ |
| GPCM | 0.213 | 0.156 | 0.186 | 0.159 | 0.163 | $-1.754$ | $-1.761$ | $-1.764$ | $-1.750$ | $-1.756$ |
| SRM | 0.243 | 0.134 | 0.179 | 0.125 | 0.155 | $-1.784$ | $-1.784$ | $-1.783$ | $-1.780$ | $-1.767$ |
| GPDM | 0.268 | 0.272 | 0.266 | 0.268 | 0.263 | $-2.155$ | $-2.158$ | $-2.158$ | $-2.159$ | $-2.158$ |
| LSM | 0.188 | 0.114 | 0.110 | 0.105 | 0.104 | $-1.997$ | $-1.960$ | $-1.908$ | $-1.845$ | $-1.775$ |
| **IPGP** | **0.322** | **0.319** | *0.323* | **0.318** | **0.318** | **$-1.478$** | **$-1.477$** | **$-1.477$** | **$-1.477$** | *$-1.476$* |

of the cross-sectional observations to learn a vanilla population factor loading matrix. We then initialize $\mathbf{W}_{\text{pop}}$ in our model with this vanilla loading matrix, and optimize the loading matrix jointly with the variational parameters. Note that $T = 1$ in LOOPR, so we drop the idiographic components.

**Validation of Big Five.** Table 3 presents the predictive accuracy and averaged log-likelihood for our method and various baselines (excluding TVAR due to its lack of low-rank assumption) across different values of $D$ in LOOPR. Bold numbers indicate the best model for each $D$, while italic numbers highlight the best model across all $D$ values. Although IPGP with $D = 5$ shows slightly lower in-sample predictive accuracy compared to the $D = 3$ model, it demonstrates significantly stronger model evidence than all alternatives. Posterior odds ratio test reveals that the second-best model is $\exp(-79) \approx 5 \times 10^{-35}$ times less likely than the five-factor model.

We further evaluated IPGP's performance through exploratory analysis, testing model ranks from 1 to 10. Results shown in Table 4 provide strong support for a rank-5 model, which achieves both higher model evidence and lower BIC, strengthening the evidence for the Big Five theory. The BIC follows a V-shaped pattern, decreasing as the rank approaches 5 and increasing thereafter, indicating that rank-5 represents an optimal balance point: ranks below 5 provide insufficient model capacity, while higher ranks lead to overfitting. These findings demonstrate that when analyzing psychological measurements from standard cross-sectional data, IPGP successfully identifies the correct underlying factor structure, making it valuable for downstream applications.

Table 4: Model performance of IPGP with model ranks from 1 to 10 in LOOPR data.

| RANK | 1 | 2 | 3 | 4 | 5 | 6 | 7 | 8 | 9 | 10 |
|---|---|---|---|---|---|---|---|---|---|---|
| **LL/N** ↑ | $-1.478$ | $-1.477$ | $-1.477$ | $-1.477$ | $\mathbf{-1.476}$ | $-1.477$ | $-1.477$ | $-1.477$ | $-1.478$ | $-1.477$ |
| **BIC** ($\times 10^{11}$) ↓ | 1.2736 | 1.2726 | 1.2728 | 1.2726 | **1.2722** | 1.2725 | 1.2726 | 1.2732 | 1.2732 | 1.2724 |

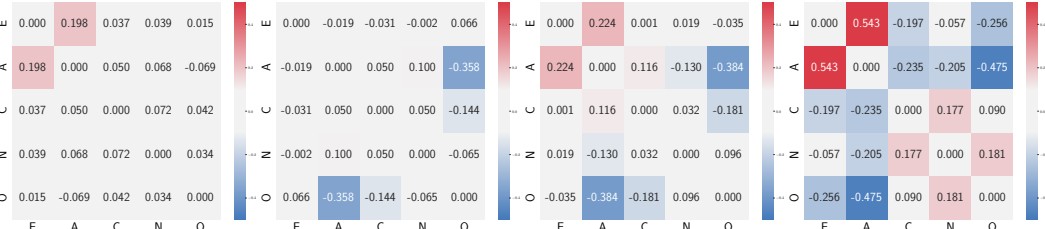

Figure 3: Four residual correlations as identified by our k-mean clustering. Each heatmap displays the trait-level residual correlation averaged across corresponding batteries for one cluster, with darker red and blue indicating larger positive and negative deviations. For instance, agreeableness (A) is more correlated to extraversion (E) than the population profile in the first profile, but less correlated to openness (O) in the second profile. Moreover, these two directions of deviations are even exacerbated in the third and fourth profiles.

**Estimated interpersonal variation.** Figure 2 compares the raw correlation matrix with our estimated Big Five correlation matrix. Both matrices exhibit a distinctive *block* pattern, characterized by strong correlations between questions within each Big Five factor and weak correlations across different factors. We also observe that questions related to negative emotionality demonstrate slight negative correlations with questions measuring extraversion and conscientiousness, suggesting the presence of meaningful trait-by-trait interaction effects.

### 4.3 Longnitudinal Pilot Study

To further demonstrate IPGP in longitudinal setting for learning idiographic psychological taxonomies, we collected intensive longitudinal data using experience sampling measures (ESM). We highlight the predictive ability of IPGP through a prediction and a leave-one-trait-out cross-validation task. We also illustrate how IPGP identifies unique personality taxonomies that might advance individualized approaches to psychological diagnosis and inspire new theory.

**Data collection.** We employed an experience sampling method (ESM) design where participants completed personality assessments six times daily over a three-week period, allowing for a maximum of 126 assessments per person. Our study included 93 valid student participants, yielding 8,770 total assessments with an average of 94 assessments per participant. We based our assessment on the BFI-2 [52], which provides comprehensive coverage of the trait space and ensures proper identification of latent factors. While the original BFI-2 contains 60 items (four items for each of the three sub-factors within each Big-Five domain), we modified it for our ESM design by removing items unsuitable for contextualized assessment. To reduce participant fatigue and minimize learning effects from repeated measures, we implemented a planned missing design: participants randomly responded to two out of three items for each sub-factor, resulting in a streamlined 30-item assessment. Note that our data is collected from a student sub-population as non-representative samples, and future studies may explore the model's applicability across diverse populations.

**Comparison between nomothetic and idiographic models.** We run the full IPGP model with idiographic component and unit-specific time kernel on the collected longitudinal data. Again we set the ranks of the population and individual loading matrices to 5 and 1 respectively, and incorporate prior knowledge of the cross-sectional data by fixing the population loadings as the Big Five loadings estimated in Sec. (4.2) and optimizing the individual loadings. We contrast our proposed idiographic model (IPGP) and baselines in Table 5, which shows the in-sample prediction, averaged log-likelihood and posterior odds ratio. We found that IPGP outperforms IPGP-NOM with higher predictive accuracy and log-likelihood, and is decisively favored by a posterior odds ratio of $\exp(1.06 \times 10^4)$.

Table 5: In-sample prediction and averaged log-likelihood of our proposed model (IPGP) and baselines for the longitudinal data, as well as log posterior odds ratios to IPGP. "—" indicates self comparison.

|  | ACC | LL/N | $\log(\textbf{OR})$ |
|---|---|---|---|
| GRM | 0.210 | $-2.266$ | $-2.32 \times 10^5$ |
| GPCM | 0.288 | $-1.516$ | $-3.80 \times 10^4$ |
| SRM | 0.260 | $-1.927$ | $-1.44 \times 10^5$ |
| GPDM | 0.382 | $-3.865$ | $-7.80 \times 10^5$ |
| LSM | 0.226 | $-1.399$ | $-7.72 \times 10^3$ |
| TVAR | 0.382 | $-1.546$ | $-4.47 \times 10^4$ |
| IPGP-NOM | 0.403 | $-1.410$ | $-1.06 \times 10^4$ |
| **IPGP** | **0.417** | $\mathbf{-1.369}$ | — |

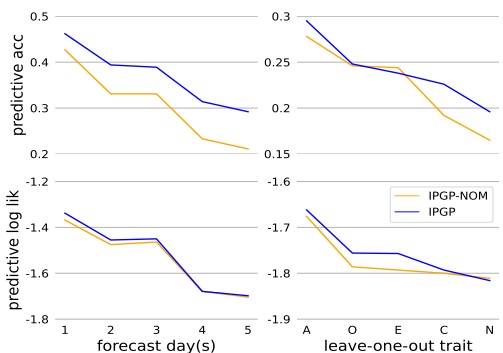

Figure 4: Predictive accuracy and log lik of IPGP and IPGP-NOM for the forecasting task and leave-one-trait-out cross-validation task.

**Predictive performance of IPGP.** We also evaluate the out-of-sample performance of the idiographic and nomothetic models using two prediction tasks: forecasting future responses and leave-one-trait-out cross validation. For the forecasting task, we train both models with data from the first 40 days and predict future responses for the last 5 days. For the cross-validation task, we predict responses of each trait by training on data belonging to the other four traits, where 20% of responses for one trait was held out (randomly choosing the trait and items to remove). Figure (4) shows the predictive accuracy and log-likelihood of IPGP and IPGP-NOM for the forecasting task over varying horizons and for the leave-one-trait-out cross-validation task. IPGP has consistently better performance than IPGP-NOM in both tasks, except for being slightly less accurate in predicting extraversion. Overall, IPGP is favored than IPGP-NOM by posterior odds ratios of $\exp(43)$ and $\exp(716)$ in these two tasks.

**Discovery of taxonomies.** Despite our small cohort size (93 respondents), we also explore how the profiles of personality that substantially differ from the interpersonal commonality cluster into informative groups. Specifically, we first perform a $k$-mean clustering using all 93 estimated individual correlation matrix with CMD as the distance metric, and then compute the residual correlation between each estimated clustering centroid and the population correlation. Figure (3) illustrates four residual correlations identified by our $k$-mean clustering. Each heatmap displays the residual correlation at the trait level averaged between the corresponding batteries for one cluster, with darker red and blue indicating larger positive and negative deviations. For instance, agreeableness (A) is more correlated to extraversion (E) than population profile in the first profile, but less correlated to openness (O) in the second profile. Moreover, these two directions of deviations are even exacerbated in the third and fourth profiles.

The discovery of unique personality taxonomies suggests a potential resolution to the longstanding idiographic versus nomothetic debate in personality and psychometric sciences. Our findings indicate that the optimal approach lies between these two extremes, rather than fully embracing either perspective. The four distinct profiles that we identified, while derived from the Big Five framework, demonstrate how individuals can meaningfully deviate from a common taxonomy. These deviations may provide valuable insights into individuals' motivations, behavioral patterns, and self-concepts. For example, individuals matching Profile 4 show a strong correlation between Extraversion and Agreeableness, possibly reflecting a tendency toward warm and socially engaging behavior (such as someone who naturally connects with and shows kindness to everyone at social gatherings). Furthermore, these distinct profiles enhance the predictive power of individual-level ($N = 1$) models by allowing them to learn from people with similar characteristic patterns.

## 5 Related work

**Idiographic assessment** captures critical individual characteristics that are often lost in simplified taxonomies of psychological behaviors [53]. Evidence from multiple psychometric fields demonstrates that nomothetic models, which focus solely on interpersonal variation, often lack generalizability

[1]. Researchers have proposed various solutions to address this limitation. Song and Ferrer [54] enhanced dynamic factor models with random effects to analyze psychological processes. Jongerling et al. [55] developed a multilevel first-order autoregressive model incorporating random intercepts to track daily positive effects across weeks. Beltz et al. [19] bridged nomothetic and idiographic approaches by introducing individual components to the group iterative multiple model (GIMME) for clinical data analysis. However, these methods share a common limitation: they model in response space rather than latent space when handling ordinal survey data.

**Gaussian process latent variable model** (GPLVM) is a dimensional reduction method for Gaussian data, where the latent variables are optimized after integrating out the function mappings [20, 56]. Our proposed framework differs from GPLVM as we optimize the factor loading matrix while marginalizing the latent variables. In addition, our model contrasts GPLVM and (variational) Gaussian process dynamical model (GPDM) [16, 17] in our non-Gaussian ordered logistic observation model and the use of multi-task kernel for latent structure. Finally, our longitudinal framework with stochastic variational inference learning differs from the static GP item response theory (GPIRT) [23] with more computationally demanding Gibbs sampling.

**Longitudinal measurement models** have gained prominence as researchers increasingly incorporate temporal dynamics into psychological theories through longitudinal survey designs [57, 58]. This development has spawned various methodological approaches. One family of methods includes longitudinal structural equation models (SEM), such as multiple-group longitudinal SEM and longitudinal growth curve models, designed for repeated measurement studies [11]. The M*plus* software later extended these capabilities by implementing dynamic SEM with Bayesian Gibbs sampling [13, 49]. Another stream of research produced dynamic item response models [9, 10, 3] and time-varying vector autoregressive models [14, 15] to track latent trait trajectories. While multi-task Gaussian process time series have been successfully applied to Gaussian observations in behavioral research [18] and found applications in causal inference [24, 26], environmental science [28], and biomedical research [30], they remain unexplored for survey batteries with non-Gaussian likelihood where exact inference is not possible.

## 6    Conclusion

We introduce the idiographic personality Gaussian process (IPGP) model, a novel approach for personalized psychological assessment that learns intrapersonal taxonomies from longitudinal ordinal survey data, a configuration that remains underexplored in Gaussian process dynamic systems literature. Our model leverages Gaussian process coregionalization to capture the between-battery structure and employs stochastic variational inference to ensure scalable computation. Looking ahead, we envision extending IPGP to other psychological domains, such as emotion research, and enhancing it by incorporating contextual information about behaviors and activities.

Our proposed IPGP framework also provides insight for domain theory testing, contributing to the substantive debate in psychometrics surrounding the shared versus unique structures of psychological features. Our experimental results show that IPGP is decisively favored over the nomothetic baseline, and substantive deviations from the common trend persist in for many individuals.

## Acknowledgments and Disclosure of Funding

This work was supported by the 2023 Seed Grant of Transdisciplinary Institute in Applied Data Sciences at Washington University in St Louis. YC and RG were supported by the National Science Foundation (NSF) under award number IIS–1845434.

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

## A  Notations of IPGP

Throughout our notation, superscript $(i)$ indicates unit and underscript $j$ indicates task.

- $\mathbf{y}^{(i)} = [\mathbf{y_1}^{(i)}, \dots, \mathbf{y_J}^{(i)}]^T$ represents the flattened $JT \times 1$ response vector consisting of all $J$ tasks and $T$ periods for unit $i$.

- $\mathbf{f}^{(i)} = [\mathbf{f_1}^{(i)}, \dots, \mathbf{f_J}^{(i)}]^T$ represents the flattened $JT \times 1$ latent vector consisting of all $J$ tasks and $T$ periods for unit $i$, mapped to response space via ordered-probit likelihood.

- $\mathbf{W}_{\text{pop}}$ is the $D \times J$ ($D$ latent dimensions, $J$ tasks) shared interpersonal loading matrix.

- $\mathbf{Z}^{(i)}$ is the $D^*$ by $J$ unit-specific low-rank loading matrix that serves to be the additional idiographic component, independent of $\mathbf{W}_{\text{pop}}$.

- $\mathbf{K}_{\text{task}}^{(i)}$ is the unit-specific task covariance matrix with shared component $\mathbf{W}_{\text{pop}}^T \mathbf{W}_{\text{pop}}$ and a low-rank approximation $\mathbf{Z}^{(i)^T} \mathbf{Z}^{(i)}$ of $D^* < D$ for unit-specific deviations.

- $\mathbf{K}_{\text{time}}^{(i)}$ is the time covariance for dynamic changes in the latent attributes. We used an RBF kernel $\mathbf{K}_{\text{time}}^{(i)}(t, t') = \exp\left( -(t - t')^2/\ell_i^2 \right)$ with unit-specific bandwidth $\ell_i$s.

$\mathbf{W}_{\text{pop}}$ is a global parameter estimated for the entire population while $\mathbf{Z}^{(i)}$ is a unit-level parameter. These are combined in Eq.(3) to create a unique kernel for each unit that combines both components.

## B  Mathematical Details of Evidence Lower Bound

We provide the full mathematical details of the evidence lower bound defined in Eq. (6). As KL divergence is always non-negative, we first consider the KL divergence between $p(\mathbf{f} \mid \mathbf{u})$ and $p(\mathbf{f} \mid \mathbf{y})$:

$$\text{KL}[p(\mathbf{f} \mid \mathbf{u}) \parallel p(\mathbf{f} \mid \mathbf{y})] = \mathbb{E}_{p(\mathbf{f}\mid\mathbf{u})} \log \frac{p(\mathbf{f} \mid \mathbf{u})}{p(\mathbf{f} \mid \mathbf{y})} \tag{7}$$

$$= \mathbb{E}_{p(\mathbf{f}\mid\mathbf{u})} \log \frac{p(\mathbf{f} \mid \mathbf{u})p(\mathbf{y})}{p(\mathbf{y} \mid \mathbf{f})p(\mathbf{f})} \tag{8}$$

$$= \mathbb{E}_{p(\mathbf{f}\mid\mathbf{u})} \log \frac{p(\mathbf{f} \mid \mathbf{u})p(\mathbf{y} \mid \mathbf{u})p(\mathbf{u})}{p(\mathbf{y} \mid \mathbf{f})p(\mathbf{f})} \tag{9}$$

$$= \mathbb{E}_{p(\mathbf{f}\mid\mathbf{u})} \log \frac{p(\mathbf{y} \mid \mathbf{u})}{p(\mathbf{y} \mid \mathbf{f})} \tag{10}$$

$$= \log p(\mathbf{y} \mid \mathbf{u}) - \mathbb{E}_{p(\mathbf{f}\mid\mathbf{u})} \log p(\mathbf{y} \mid \mathbf{f}) \geq 0 \tag{11}$$

Moving $\mathbb{E}_{p(\mathbf{f}\mid\mathbf{u})} \log p(\mathbf{y} \mid \mathbf{f})$ to the R.H.S of the above inequality will lead to Eq. (4). We then exploit the inequality given by $\text{KL}[q(\mathbf{u}) \parallel p(\mathbf{u} \mid \mathbf{y})] \geq 0$:

$$\text{KL}[q(\mathbf{u}) \parallel p(\mathbf{u} \mid \mathbf{y})] = \mathbb{E}_{q(\mathbf{u})} \log \frac{q(\mathbf{u})}{p(\mathbf{u} \mid \mathbf{y})} \tag{12}$$

$$= \mathbb{E}_{q(\mathbf{u})} \log \frac{q(\mathbf{u})p(\mathbf{y})}{p(\mathbf{y} \mid \mathbf{u})p(\mathbf{u})} \tag{13}$$

$$= -\mathbb{E}_{q(\mathbf{u})} \log p(\mathbf{y} \mid \mathbf{u}) + \text{KL}[q(\mathbf{u}) \parallel p(\mathbf{u})] + \log p(\mathbf{y}) \geq 0 \tag{14}$$

Rearranging the above inequality, applying Eq. (4) and exploiting notation $q(\mathbf{f}) = \int p(\mathbf{f} \mid \mathbf{u})q(\mathbf{u})d\mathbf{u}$ leads to the ELBO:

$$\log p(\mathbf{y}) \geq \mathbb{E}_{q(\mathbf{u})} \log p(\mathbf{y} \mid \mathbf{u}) - \text{KL}[q(\mathbf{u}) \parallel p(\mathbf{u})] \tag{15}$$

$$= \mathbb{E}_{q(\mathbf{u})}\big[\mathbb{E}_{p(\mathbf{f}\mid\mathbf{u})} \log p(\mathbf{y} \mid \mathbf{f})\big] - \text{KL}[q(\mathbf{u}) \parallel p(\mathbf{u})] \tag{16}$$

$$= \mathbb{E}_{q(\mathbf{f})} \log p(\mathbf{y} \mid \mathbf{f}) - \text{KL}[q(\mathbf{u}) \parallel p(\mathbf{u})] \tag{17}$$

## C  Data Collection and Demographics

The LOOPR dataset from Soto [50] in our first case study used Qualtrics and quota sampling to ensure representative samples of the U.S. population, and hence was very diverse:

- age: 11% ages 18-24, 18% ages 25-34, 17% ages 35-44, 19% ages 45-54, 17% ages 55-64, 18% ages 65 and older
- sex: 52% female, 48% male
- race/ethnicity: 74% non-Hispanic white/Caucasian, 11% black/African American, 10% Hispanic/Latino, 3% Asian/Asian American, 2% American Indian/Native American
- educational attainment: 10% did not complete high school, 33% high school graduate, 28% some college, 19% college graduate, 10% graduate or professional degree
- annual household income ($): 14% <20,000, 12% 20,000-29,999, 11% 30,000-39,999, 15% 40,000-49,999, 26% 50,000-79,999, 22% 80,000+

As social and personality psychology often faces challenges with model building due to reliance on non-representative samples, past researches rely heavily on student populations. We follow such practice but aim to broaden our sample within budget and available diversity. Participants in our longitudinal study (the third experiment) are mostly college students with an average age of 20.23 (SD = 1.94); 70% are female, 26% are male and 4% are self-identified as other; ethnicities self-reported as 42% Caucasian, 39% Asian, 12% African American, and 7% Other. We used the BFI-2 [52], a widely used personality measure across cultures and ages. We acknowledge the typical biases of convenient sampling in higher education, including socio-economic and ethnic diversity limitations.

## D  Runtime in Simulation Study

Table 6 shows the average runtime of IPGP and its competitors in the simulation study. IPGP does require more time to train due to the enlarged model space.

Table 6: Average runtime of IPGP and its competitors in the simulation study.

| MODEL | AVG RUNTIME (SEC) |
|---|---|
| GRM | 343 |
| GPCM | 367 |
| SRM | 1398 |
| GPDM | 17359 |
| LSM | 311 |
| TVAR | 468 |
| IPGP-NOM | 17594 |
| IPGP-IND | 21562 |
| IPGP-LOW | 10839 |
| IPGP-NP | 31150 |
| IPGP | 30141 |

# E Estimated Correlations of Selective Individuals

Figure (5) shows the estimated correlations of selective individuals for the identified four profiles in the longitudinal study.

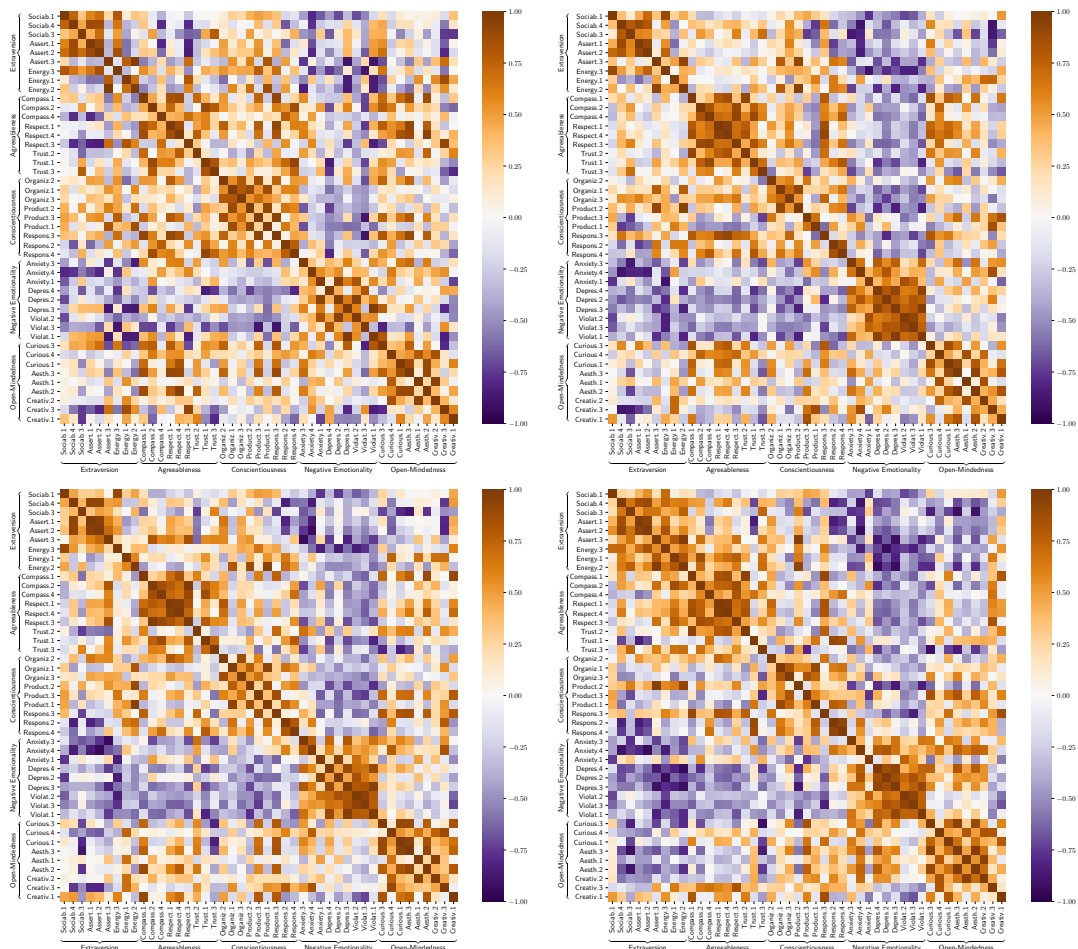

Figure 5: Estimated correlations of selective individuals for the identified four profiles in the longitudinal study.

