# OpenReview forum: "Idiographic Personality Gaussian Process for Psychological Assessment"
_NeurIPS.cc/2024/Conference — NeurIPS 2024 poster_

### Official Review · Reviewer_A4W7 · 2024-07-08

**Soundness:** 4
**Presentation:** 4
**Contribution:** 3
**Rating:** 7
**Confidence:** 4

**Summary:**

This paper presents a multi-output Gaussian process model for classification in the context of psychological studies. It's based on a linear model of co-regionalisation leveraging unit-level latent factors and RBF kernels to jointly model population and individual personality traits to tackle an old debate in psychometrics regarding their original nature. The authors leverage a variational formulation to derive a lower bound for the model evidence and optimise the individual-specific, population, and GP-related parameters. An extensive simulation and real-data study is proposed, presenting longitudinal and multi-subject survey analyses, personality correlations and predictions.

**Strengths:**

This paper is remarkably well-written and presented. The flow of derivations is easy to follow, and the illustrations are helpful and of excellent quality. The treated problem is far from trivial, with data that present several degrees of correlations (time, individual, population) and technical constraints (missing and categorical data). The experiments are impressive, with extensive comparison against many competitors and excellent results overall. The method shows great promise in providing practical insight into the domain, and even though I'm not a specialist in psychology, the ability to reconcile idiographic and nomothetic approaches seems particularly valuable.

**Weaknesses:**

One could argue that the methodological novelty is limited as the presented multi-output GP model is fairly well-known in the GP community. However, I think its similarities and differences with GPLVM and GPDM are well presented, and the overall application is far from trivial regarding the nature of data (categorical outputs, latent variables, missing data, ...). This relative weakness is more than compensated by the strength of results and the potential such a methodology brings in a field like psychology, where the signal-on-noise ratio is generally low, and the nature of measurements is often challenging.

**Questions:**

I only have a few questions regarding computation times, scaling and robustness of the inter-output covariance matrices, which are well-known limitations of co-regionalisation GP models in practice.

- In this sense, could you provide numerical evidence and comparison regarding training and prediction times for IPGP and competitors?
- Even though the number of tasks/topics remains probably limited in psychological applications, could you discuss a bit more, in Section 4.1 **Result**, the decrease in performance between full IPGP and the low-rank version. What could we expect for higher dimensions? Would the low-rank approximation remain robust for a more significant gap with the actual rank?
- In my experience, several multi-output GP models (or maybe their implementations) are somewhat unstable when it comes to estimating $\textbf{K}_{task}$, and pathological cases can often arise. Have you experienced such problems? And if so, how did you manage to tackle this issue?

**Typos:**

Page 5: The indices of the CMD definition have a slight error. All terms are written as $R_1$, and the norm is 'f' instead of '$l_2$'
Page 6: The title of 4.2 "Cross-secitonal" ==> Cross-sectional
Page 7: "Both correlation matrices **displace** a block pattern" ==> display?
Page 7: "questions corresponding negative emotionality" ==> corresponding to

**Limitations:**

In my opinion, the limitations are adequately discussed. The application spectrum and position in the related literature are well documented.

---

> ### Author Rebuttal · Authors · 2024-08-06
>
> Dear Reviewer A4W7,
>
> Thank you for your valuable feedback. Please see our responses below
>
> > In this sense, could you provide numerical evidence and comparison regarding training and prediction times for IPGP and competitors?
>
> Sure, please see below. We’d however first like to comment that runtime is typically not a primary concern in such analysis. The data we collected here required several months of time and over $20k in subject payments, so a few hours (or even a couple days) of one-time work is quite reasonable in light of that initial outlay if it provides additional insight.
>
> The table below shows the average runtimes of IPGP and competitors on the simulation study data from the paper. IPGP does require more time to train due to the larger model space, but the runtime is still quite manageable even with a non-optimized reference implementation. We will add this table in the appendix and include some explicit discussion on runtime to the manuscript.
>
> | Model | GRM  | GPCM | SRM  | GPDM | DSEM | TVAR | IPGP-NOM | IPGP-IND | IPGP-LOW | IPGP-NP | IPGP  |
> |---------------|------|------|------|------|------|------|----------|----------|----------|---------|-------|
> | Avg runtime (sec)   | 343  | 367  | 1398 | 17359| 311  | 468  | 17594    | 21562    | 10839    | 31150   | 30141 |
>
> > Even though the number of tasks/topics remains probably limited in psychological applications, could you discuss a bit more, in Section 4.1 Result, the decrease in performance between full IPGP and the low-rank version. What could we expect for higher dimensions? Would the low-rank approximation remain robust for a more significant gap with the actual rank?
>
> In general, a low-rank approximation inevitably lacks the model capacity of its full-rank counterpart, but this deficiency only matters up to the intrinsic rank of the data. Blindly increasing the rank beyond that point will not improve performance. In practice, when the rank is not known, a data-driven search could be performed to identify a suitable rank.
>
> We ran an additional experiment during the rebuttal period using the simulation data from Section 4.1 in the manuscript where we varied the model rank in {2, 5, 8} (recall the true rank of the data was 5), repeating this simulation using 25 different random seeds. The results are summarized in the below table. Both the true rank (5) and high rank (8) models outperform the low rank (2) model, but there is no performance gain by increasing the rank more than necessary.
>
> We will add this table and discussion to the appendix.
>
> | Rank | Train Acc          | Train LL            | Test Acc           | Test LL            | CMD               |
> |------|--------|------------|-----------|-------|-------|
> | 2    | 0.897 $\pm$ 0.004  | −0.313 $\pm$ 0.010  | 0.884 $\pm$ 0.005  | −0.334 $\pm$ 0.011 | 0.397 $\pm$ 0.007 |
> | 5    | 0.957 $\pm$ 0.002  | −0.159 $\pm$ 0.005  | 0.942 $\pm$ 0.002  | −0.184 $\pm$ 0.006 | 0.128 $\pm$ 0.006 |
> | 8    | 0.957 $\pm$ 0.002  | −0.161 $\pm$ 0.004  | 0.945 $\pm$ 0.002  | −0.183 $\pm$ 0.005 | 0.124 $\pm$ 0.006 |
>
>
> > In my experience, several multi-output GP models (or maybe their implementations) are somewhat unstable when it comes to estimating, and pathological cases can often arise. Have you experienced such problems? And if so, how did you manage to tackle this issue?
>
> We have not experienced such issues. The GPyTorch software that our multi-output GP implementation is based on is quite stable.
>
> > Typos:
>
> Thank you for pointing these out. We will fix them in revision.

---

> ### Comment · Reviewer_A4W7 · 2024-08-13
>
> Thank you for your thorough answers. I understand the technical constraints coming from such costly and time-consuming studies involving human subjects. While I acknowledge that this discussion about running time might be superfluous for your application, I always prefer to see beforehand 'how much exactly' it will cost me when I intend to test a method on my data, and I suspect that future readers would, too. This is especially true for methods like MTGP, which I know are coming at a not negligible cost.
>
> That being said, I commend the authors' efforts to provide additional evidence and thus improve the confidence one can grant this paper. More generally, I support the authors' view about 'building bridges' between ML methodologies and their applications. Although I consider myself more of a methodological researcher, I notice that we too often focus on 'novelty' by principle while neglecting meticulous and well-conducted applications. Such studies are crucial for advancing more experimental disciplines like psychology with all the rigour it deserves from ML researchers. We all agree that this paper aims not to *advance ML* vastly, but I appreciate that it *uses ML to advance*.

---

> > ### Author Response · Authors · 2024-08-13
> >
> > > I always prefer to see beforehand 'how much exactly' it will cost me when I intend to test a method on my data, and I suspect that future readers would, too. This is especially true for methods like MTGP, which I know are coming at a not negligible cost.
> >
> > This view resonates with us. We're more than happy to add this discussion and these results regarding running time to the paper to help complete the story. We appreciate your original comment and the invitation to run these additional experiments.
> >
> > > That being said, I commend the authors' efforts to provide additional evidence and thus improve the confidence one can grant this paper. More generally, I support the authors' view about 'building bridges' between ML methodologies and their applications. Although I consider myself more of a methodological researcher, I notice that we too often focus on 'novelty' by principle while neglecting meticulous and well-conducted applications. Such studies are crucial for advancing more experimental disciplines like psychology with all the rigour it deserves from ML researchers. We all agree that this paper aims not to advance ML vastly, but I appreciate that it uses ML to advance.
> >
> > We truly appreciate your support!

---

### Official Review · Reviewer_mvMV · 2024-07-10

**Soundness:** 3
**Presentation:** 3
**Contribution:** 2
**Rating:** 6
**Confidence:** 5

**Summary:**

Gives a multi-task/output GP formulation for multiple time-series (or intrinsic co-regionalisation model) for pyschological assessments. Design of the factor loadings informs the task-correlations and reflects the knowledge of individual's correlation between his responses and inter-person correlations. More of an application paper rather than a paper advancing machine learning; but a good application nonetheless.

**Strengths:**

1. The application is a natural fit for the model chosen, and results are good and convincing.
2. The use of Bayes factor for model testing make the paper better as an reference for this application.
3. Sections 4.2 and 4.3 evaluates the model from different aspects.

**Weaknesses:**

1. The paper is weak from the perspective of advancing state-of-the-art machine learning algorithms.
2. The paper is totally unrelated to GPLVM and tasks that GPLVM are designed for. Mentions of GPLVM only confuses the reader.
3. Equation 3 needs fixing.
4. In section 3.2, the method is variational inference, and not stochastic variational inference.
5. Description of the setup in section 4.1 needs to be clearer to explain also in terms of $K_{task}$.
6. Line 192 mentions "informative prior". What is this "informative prior"?

*Minor*
7. Line 319: "addressing" is too strong. Suggest to change to "contributing".
8. Line 321: "than" -> "over"

**Questions:**

I do not have any critical questions.

**Limitations:**

Yes.

---

> ### Author Rebuttal · Authors · 2024-08-05
>
> Dear Reviewer mvMV,
>
> Thank you for your valuable feedback. Please see our responses below.
>
> > The paper is weak from the perspective of advancing state-of-the-art machine learning algorithms.
>
> Please see the shared rebuttal regarding the scope  and nature of our contributions.
>
> > The paper is totally unrelated to GPLVM and tasks that GPLVM are designed for. Mentions of GPLVM only confuses the reader.
>
> Please allow us to elaborate on this claim. We believe that IPGP is spiritually related to GPLVM in that both involve the estimation of low-dimensional latent variables in order to gain more insight into high-dimensional observations. We agree that there is considerable departure in how that estimation proceeds. Further, reviewer A4W7 commented on the necessity of including GPLVM in the related work: “I think its similarities and differences with GPLVM and GPDM are well presented, and the overall application is far from trivial regarding the nature of data”.
>
> > Equation 3 needs fixing.
>
> Indeed, there is a misplaced transpose; it should be $W'W + ww'$, where $W$ is a $K\times J$ matrix and $w$ is a $J\times 1$ column vector. We will update accordingly. (Please indicate if you had something else in mind.)
>
> > In section 3.2, the method is variational inference, and not stochastic variational inference.
>
> Thank you for pointing this out. We will revise accordingly.
>
> > Description of the setup in section 4.1 needs to be clearer to explain also in terms of $W_\text{task}$.
>
> Thank you for pointing this out. In Section 4.1 we describe the setup of the shared interpersonal loading matrix $W_\text{pop}$ and unit-specific loading $w_i$, which are then used to construct the $W_\text{task}$ matrix according to Equation 3. We will make this more explicit in revision.
>
> > Line 192 mentions "informative prior". What is this "informative prior"?
>
> The informative prior refers to the interpersonal loading matrix $W_\text{pop}$ estimated from the standard cross-sectional data (the LOOPR data), which is used to reduce the number of hyperparameters of the full idiographic kernel from K*J (J items, K dimensions) + n*J (J items, n individuals) to nJ. Practically, the model training procedure includes two steps: (1) learning $W_\text{pop}$ using cross-sectional data, and then (2) learning the rest of the hyperparameters while holding $W_\text{pop}$ fixed. We show in the simulation that IPGP achieves more precise estimation of individual taxonomies with this stronger prior (see Table 1).
>
> > (Minor) Line 319: "addressing" is too strong. Suggest to change to "contributing". Line 321: "than" -> "over"
>
> Thank you for the suggestions; we will adopt them in revision.

---

> > ### Comment · Reviewer_mvMV · 2024-08-10
> >
> > I think it is a matter of interpreting "bridge". It is clearly a "bridge" in terms of bringing new applications in. It is not a "bridge" in terms of bringing new ideas to advance ML --- an example of which is Random Matrix Theory.
> >
> > For GPLVM, I am agreeable to it being mentioned in related work. Claiming in the introduction that you "advances on the  ... GPLVM" is simply too much for me to take.
> >
> > I may reconsider the score during the reviewer discussion stage.

---

> > > ### Author Response · Authors · 2024-08-10
> > >
> > > Thank you for your response!
> > >
> > > > Claiming in the introduction that you "advances on the ... GPLVM" is simply too much for me to take.
> > >
> > > We are happy to rephrase this passage, in particular to avoid the acronym GPLVM entirely. We intended to indicate that our model was among a family of spiritually related models incorporating latent variables and Gaussian processes in their construction (it is "a" latent variable GP model), not that we were advancing "the" famous GPLVM model from Lawrence.

---

### Official Review · Reviewer_YTHZ · 2024-07-12

**Soundness:** 2
**Presentation:** 3
**Contribution:** 2
**Rating:** 3
**Confidence:** 5

**Summary:**

1. This paper introduces an innovative measurement framework utilizing the Gaussian process coregionalization model to resolve the question of whether psychological attributes such as personality exhibit a universal structure among the populace or are uniquely individualized.
2. An Idiographic Personality Gaussian Process (IPGP), a hybrid model that accounts for both the commonality of traits across people and individual-specific "idiographic" variations.
3. Gaussian process coregionalization model to interpret the responses from grouped survey batteries, adapted for non-Gaussian ordinal data, and employs stochastic variational inference for estimating latent factors.
4. The application of IPGP on both synthetic data and an original survey demonstrates its performance.

**Strengths:**

1. The paper’s main innovation lies in its unique combination of multitask Gaussian process, which results in a conceptualization of the subject matter.
2. The innovative use of multitask Gaussian process in this paper offers a promising solution to a long-standing challenge of psychological assessment.
3. The quantitative and qualitative methods in this paper provide a potential in advancing psychological diagnosis and treatment.
4. The interdisciplinary nature of the research is interesting.
5. The authors provide a holistic view of the issue at hand.

**Weaknesses:**

1. there are many MTGPs, the author ignored comparison with them.
2. incorrect description, such as: "multi-task structure is also known as the linear model of coregionalization (LMC)".
3. This represents a particular implementation of MTGP, but it doesn't introduce any novel methodological advancements.
4. The paper presents a valuable contribution, but its innovative aspects are limited, as it largely builds upon existing theories without introducing significant new insights.
5. The literature review appears somewhat limited. Expanding it to include more recent or relevant MTGPs could provide a more comprehensive context for the research.
6. The statistical analysis used in the study seems inadequate given the complexity of the data.
7. The research seems to be more of an incremental advancement rather than a novel contribution, which may limit its impact on the field.
8. The experimental evidence provided in the paper is not as comprehensive as it should be to support the claims made, suggesting a need for more extensive testing.
9. Only one experiment is related to psychological assessment.
10. No idiographic personality is discovered and presented in the paper.

**Questions:**

Please see the weakness above

**Limitations:**

Please see the weakness above

---

> ### Author Rebuttal · Authors · 2024-08-05
>
> Dear Reviewer YTHZ,
>
> Thank you for your valuable feedback. Please see our responses below.
>
> > there are many MTGPs, the author ignored comparison with them
>
> To our knowledge, Duerichen et al. [17] is the only existing MTGP model in the behavioral literature for multivariate physiological time-series analysis, but focuses on modeling responses directly rather than uncovering the latent structure, as we do here. So the construction of additional MTGPs for this setting is a novel contribution of its own. Note that we do include several ablated models in Tables 1-3 that are variants of MTGPs.
>
> > incorrect description, such as: "multi-task structure is also known as the linear model of coregionalization (LMC)".
>
> Indeed, the linear model of coregionalizaiton is only one (quite common) realization of the multi-task GP framework. We will reword accordingly.
>
> > This represents a particular implementation of MTGP, but it doesn't introduce any novel methodological advancements.
>
> Please see the shared rebuttal regarding the scope  and nature of our contributions.
>
> > The paper presents a valuable contribution, but its innovative aspects are limited, as it largely builds upon existing theories without introducing significant new insights.
>
> Please see the shared rebuttal regarding the scope  and nature of our contributions.
>
>
> > The literature review appears somewhat limited. Expanding it to include more recent or relevant MTGPs could provide a more comprehensive context for the research.
>
> MTGPs are underexplored in the behavioral literature. We did include a discussion on existing MTGP models for multivariate physiological time-series analysis [Duerichen, et al]. Outside the field of psychology, MTGPs have been recently applied in causal inference [1,2,3], environmental science [4, 5, 6] and biomedical research [7]. We add these related works to the literature review.
>
> [1] Aglietti et al. NeurIPS 2020
>
> [2] Alaa and van der Schaar. NeurIPS 2017
>
> [3] Chen, et al. AISTATS 2023
>
> [4] Zhou, et al. Journal of Cleaner Production doi:10.1016/j.jclepro.2020.124710
>
> [5] Li, et al. Measurement doi:10.1016/j.measurement.2021.110085
>
> [6] Dahl and Bonilla. Machine Learning doi:0.1007/s10994-019-05808-z
>
> [7] Zhang, et al. Journal of Biomedical Informatics doi:10.1016/j.jbi.2022.104079
>
> > The statistical analysis used in the study seems inadequate given the complexity of the data.
>
> We used paired t-tests for model evaluation in Table 1 and likelihood-ratio tests in Tables 2-3, both of which are standard tools for statistical hypothesis testing. We are happy to continue this conversation. Could you please provide more details regarding your concern (for example, an alternative approach or a specific deficiency in ours) so that we can offer meaningful additional comments during the author-reviewer discussion phase?
>
> > The experimental evidence provided in the paper is not as comprehensive as it should be to support the claims made, suggesting a need for more extensive testing.
>
> We summarize our claims and empirical evidence that supports the claims here:
>
> 1. IPGP is a novel psychological assessment model that improves both prediction of actual
> responses and estimation of individualized factor structures relative to existing benchmarks (see Table 1-2 for evidence).
>
> 2. Substantive deviations from the common psychological structures persist in considerable individuals, as IPGP is decisively favored than the nomothetic baseline (see Table 3 for model comparison and Figure 4 for our learned idiographic profiles).
>
> We are happy to continue this conversation. Could you please provide more details regarding your concern (for example, an alternative approach or a specific deficiency in ours) so that we can offer meaningful additional comments during the author-reviewer discussion phase?
>
> > Only one experiment is related to psychological assessment.
>
> Our paper includes three experiments: a simulation study, a re-analysis of a large cross-sectional dataset (itself a psychological assessment), and a pilot study of repeated measures of the Big Five personality traits (a second, separate assessment). We note that collecting this pilot study was itself an intensive investment requiring multiple months and over $20k in subject payments. Repeated assessment datasets of this type are not widely shared to protect human subjects, hence why we had to collect our own.
>
> > No idiographic personality is discovered and presented in the paper.
>
> Figure 4 presents four residual correlation matrices w.r.t to the main population, each highlighting one distinct profile of idiographic personality. Selected full correlation matrices of idiographic personality types can also be found in Appendix B. Further, the results in Table 3 show clearly that the model that allows for idiographic personality structure is strongly favored.

---

> ### Comment · Reviewer_YTHZ · 2024-08-08
>
> Employing other advanced MTGPs for this psychological task is straightforward. However, the author intentionally abandoned more comparisons with other MTGPs.
>
> The innovation of the method proposed in this paper is quite limited.
>
> This article makes a very small contribution to the field of Gaussian processes
>
> Compared to state-of-the-art methods, its advancements and advantages are minimal.
>
> As this is an applied work, I suggest the authors submit their paper to a journal focused on bioinformatics.

---

> > ### Author Response · Authors · 2024-08-12
> >
> > Thank you for your response, although we respectfully disagree.
> >
> > We strongly believe work such as ours has a place at this conference.
> >
> > > As this is an applied work, I suggest the authors submit their paper to a journal focused on bioinformatics.
> >
> > The call for papers explicitly encourages submissions of applications and highlights the participation of diverse communities beyond core ML:
> >
> > > [NeurIPS] brings together researchers in machine learning, neuroscience, statistics, optimization, computer vision, natural language processing, life sciences, natural sciences, social sciences, and other adjacent fields.
> >
> > > We invite submissions presenting new and original research on topics including but not limited to the following:
> >
> > > - Applications
> > > [...]
> > > - Machine learning for sciences (e.g. climate, health, life sciences, physics, social sciences) [...]
> >
> > We do not believe it is in the spirit of the call for papers to dismiss applied work out of hand.
> >
> > > The innovation of the method proposed in this paper is quite limited.
> > > This article makes a very small contribution to the field of Gaussian processes
> >
> > The reviewer guidelines explicitly encourage diversity in contributions beyond purely methodological improvements to (MT)GPs:
> >
> > > There are many examples of contributions that warrant publication at NeurIPS.  These contributions may be theoretical, methodological, algorithmic, empirical, connecting ideas in disparate fields (“bridge papers”), or providing a critical analysis.
> >
> > Our contributions here required both considerable expertise in GP modeling (including the expertise required to build several necessary innovations highlighted by reviewer A4W7) and considerable and sustained engagement with another domain to ensure success. Our strong results reflect a sophisticated model shedding light on an important psychological question.
> >
> > We do not believe it is in the spirit of these guidelines to dismiss our non-methodological (empirical, bridging communities mentioned in the CFP, etc.) contributions out of hand -- especially as they are supported by methodological contributions as well!

---

### Official Review · Reviewer_5m3j · 2024-08-15

**Soundness:** 3
**Presentation:** 3
**Contribution:** 3
**Rating:** 6
**Confidence:** 4

**Summary:**

UPDATE: I am updating my scores in light of the excellent authors' response.

This paper considers psychometric data composed of ordinal responses $y_{ijt}$, each of which represent how unit _i_ answered survey item _j_ during time period _t_. The key characteristic of this item-response data is that the same $N$ units are longitudinally surveyed on the same $J$ items over $T$ periods.

The paper takes an ordered logit factor modeling approach to analyzing such data, wherein $f_j^{(i)}(t) = \mathbf{w}_j^\top \mathbf{x}_i(t)$ is the latent ideal point of unit _i_ on on item _i_ at time _t_, and $\mathbf{w}_j$ and $ \mathbf{x}_i(t)$ are the $K$-dimensional loadings and factor vectors, respectively.

The paper moreover advocates an "idiographic approach [which] emphasizes _intrapersonal_ variation by requiring distinct loadings $\mathbf{w}_j^{(i)}$" which are different for each unit $i$. It is able to effectively achieve this by exploiting the repeated measurements of each unit-item $(i,j)$ pair over different time periods $t$.

The proposed model is an instance of a multi-task Gaussian process (MTGP). Intrapersonal variation is modeled using a unit-specific kernel $K_{time}^{i}$. The full covariance is then $JT \times JT$, resulting from a Kronecker product of $\mathbf{K}_{\textrm{time}}^{(i)}$ (applied to input $T$ periods) with a low-rank matrix $J \times J$ matrix that represents covariance between survey items (or "tasks"). This all extends the previously-presented linear model of coregionalization (LMC), which was originally developed for the simpler case where observations of $(i,j)$ pairs are not repeated over time.

The paper derives a variational inference inference algorithm for the model, which follows closely from previous work.

The paper then reports three sets of experiments: 1) a synthetic study that reports excellent results on parameter recovery (when the true parameters are known), 2) a re-analysis study that purports to show that a non-dynamic (i.e., non-idiographic) version of proposed model is able to identify the correct factor structure (K=5) from survey data designed to assess the "Big Five" psychometric traits, and 3) an illustrative case study involving a novel longitudinal data set.

**Strengths:**

The paper is very well-written. It brings an interesting application to life and convinces the reader that the proposed modeling approach is well-tailored to the problem at hand. The paper gives a clear review of the prerequisite concepts and presents its modeling approach clearly.

The paper covers related work in psychometrics well is convincing that the modeling approach is novel within that applied community. Experiments bear out this claim, as the proposed model performs much better than models which are currently used in psychometrics.

The paper presents a novel longitudinal survey dataset composed of $N=93$ subjects who were given personality assessment surveys over the course of three weeks. If I am reading the paper correctly, each subject was asked to complete a survey _six times per day_. This data sounds highly non-trivial to collect, and should be considered a main contribution in itself.

**Weaknesses:**

The paper is vague about its technical contributions. It makes the following hedged novelty statement: "the first multi-task GP latent variables model _for dynamic idiographic assessment_". I took this to mean that the proposed approach is not very new technically, but it has never yet been applied to dynamic idiographic assessments. However, the paper also makes the general claim that it "advances the literatures on Gaussian process latent variable models...". I am of the opinion that tailoring existing modeling frameworks to new applications does provide a technical contribution, as it contributes a new view and set of interpretations/metaphors that can help to better understand the abstract model. I think the paper does contribute in this way. But I am unsure if the paper further contributes more substantially to the area of Gaussian process latent variable models, as the paper is vague about that.

There is sloppiness in some of the main equations. For instance, equation 2 confuses a distribution with a random variable stating $p(\mathbf{f}^{(i}) \sim \cdots$, and I believe equation 3 confuses an inner with an outer product (shouldn't it be $w_i w_i^\top$?). There's also some confusing overloading of symbols between the background and the model sections. The background sets up the idea that an ideographic approach has distinct loadings $w_j^{(i)}$ different across $i$. But the proposed model itself does not seem to sport this as each $w_j$ is global for each task. I believe effectively the model does achieve an idiographic interpretation through the unit-specific kernel, but the connection between the background and model section is not clear.

I think the second experiment has some major flaws. The paper claims this experiment validates the "Big Five" because model performance peaks at $K=5$. However, the model is only fit for $K=1...5$. We can clearly see from the log likelihood numbers that model performance for IPGP is monotonically increasing in $K$ for the values considered, suggesting that it will likely be even better at larger values of $K$; if that were true, it would not validate the "Big Five" theory. As this is an applied paper, I would expect a much higher level of rigor on one of the two applied case studies.

**Questions:**

As this is an emergency review, after the reviewer discussion period, I am not sure if the authors will be able to reply. But if they are able, I would just ask that they respond to the various points made in previous sub-sections.

**Limitations:**

This paper did involve a fairly intense human subjects experiment for data collection, but the paper reports it was IRB-approved. Psychometrics of this kind are fairly common; I do not think the paper introduces any new potential negative societal impacts.

---

### Author Rebuttal · Authors · 2024-08-06

Two reviewers commented on the nature of our contributions. We post a shared comment on the scope of our contributions here.

We would like to first stress that our contributions are not merely theoretical, but that our work also represents applied machine learning for science. Both applications and ML for science are listed as topics of interest in the call for papers https://neurips.cc/Conferences/2024/CallForPapers. We would also like to highlight the following language in the both the call and reviewer instructions:

> There are many examples of contributions that warrant publication at NeurIPS. These contributions may be theoretical, methodological, algorithmic, empirical, connecting ideas in disparate fields (‘bridge papers’), or providing a critical analysis.

A primary contribution of our work is in “bridging” between advances in (MT)GP modeling and the disparate field of psychological assessment and diagnosis. Building this bridge required significant engagement from another community and significant care in modeling to address the nuances and challenges presented by this setting (to quote reviewer A4W7: “categorical outputs, latent variables, missing data, ...”). Indeed, reviewer A4W7 commented further on the nature -- and relative strength -- of our contributions in this light:

> One could argue that the methodological novelty is limited as the presented multi-output GP model is fairly well-known in the GP community. However, I think its similarities and differences with GPLVM and GPDM are well presented, and the overall application is far from trivial regarding the nature of data (categorical outputs, latent variables, missing data, ...). This relative weakness is more than compensated by the strength of results and the potential such a methodology brings in a field like psychology, where the signal-on-noise ratio is generally low, and the nature of measurements is often challenging.

To provide our contributions here a bit more context, and underscore the importance of the bridge we build here, there is an underlying debate in psychology that we are addressing with this work: whether (i) all individuals have a shared personality structure, (ii) all individuals have a (unique) idiosyncratic structure, or (iii) something in between. The IPGP framework offers a powerful new tool to engage in this debate as each of these scenarios can be modeled with a special case of our model. Bayes factors (and predictive capacity) then allow us to measure the relative merit of these claims in a principled and data-driven manner.

Our experimental results reaffirm the common Big Five personality model through a factor analysis study in Table 2, and show that the idiographic model is decisively favored to the nomothetic model in Table 3. Moreover, we identify distinct personality profiles that substantially differ from the interpersonal commonality (see Figure 4). Our results therefore provide a significant challenge to dominant paradigms of personality by showing evidence for an intermediate outcome where there is neither a perfectly shared structure nor a perfectly idiosyncratic structure, but rather structured idiosyncratic deviations from a common baseline. We believe that our methodological approach sets the stage for significant advancements in theorizing, evaluation, and (eventually) clinical care in the psychological domain.

---

### Comment · Program_Chairs · 2024-08-16

Hi all,

We re-opened the discussion for this paper to Aug 17 11:59pm ET for authors to reply to the emergency review. The review is pasted below.

======

**Summary**

This paper considers psychometric data composed of ordinal responses $y_{ijt}$, each of which represent how unit _i_ answered survey item _j_ during time period _t_. The key characteristic of this item-response data is that the same $N$ units are longitudinally surveyed on the same $J$ items over $T$ periods.

The paper takes an ordered logit factor modeling approach to analyzing such data, wherein $f_j^{(i)}(t) = \mathbf{w}_j^\top \mathbf{x}_i(t)$ is the latent ideal point of unit _i_ on on item _i_ at time _t_, and $\mathbf{w}_j$ and $ \mathbf{x}_i(t)$ are the $K$-dimensional loadings and factor vectors, respectively.

The paper moreover advocates an "idiographic approach [which] emphasizes _intrapersonal_ variation by requiring distinct loadings $\mathbf{w}_j^{(i)}$" which are different for each unit $i$. It is able to effectively achieve this by exploiting the repeated measurements of each unit-item $(i,j)$ pair over different time periods $t$.

The proposed model is an instance of a multi-task Gaussian process (MTGP). Intrapersonal variation is modeled using a unit-specific kernel $K_{time}^{i}$. The full covariance is then $JT \times JT$, resulting from a Kronecker product of $\mathbf{K}_{\textrm{time}}^{(i)}$ (applied to input $T$ periods) with a low-rank matrix $J \times J$ matrix that represents covariance between survey items (or "tasks"). This all extends the previously-presented linear model of coregionalization (LMC), which was originally developed for the simpler case where observations of $(i,j)$ pairs are not repeated over time.

The paper derives a variational inference inference algorithm for the model, which follows closely from previous work.

The paper then reports three sets of experiments: 1) a synthetic study that reports excellent results on parameter recovery (when the true parameters are known), 2) a re-analysis study that purports to show that a non-dynamic (i.e., non-idiographic) version of proposed model is able to identify the correct factor structure (K=5) from survey data designed to assess the "Big Five" psychometric traits, and 3) an illustrative case study involving a novel longitudinal data set.

**Soundness**: 3: good
**Presentation**: 3: good
**Contribution**: 2: fair

**Strengths**

The paper is very well-written. It brings an interesting application to life and convinces the reader that the proposed modeling approach is well-tailored to the problem at hand. The paper gives a clear review of the prerequisite concepts and presents its modeling approach clearly.

The paper covers related work in psychometrics well is convincing that the modeling approach is novel within that applied community. Experiments bear out this claim, as the proposed model performs much better than models which are currently used in psychometrics.

The paper presents a novel longitudinal survey dataset composed of $N=93$ subjects who were given personality assessment surveys over the course of three weeks. If I am reading the paper correctly, each subject was asked to complete a survey _six times per day_. This data sounds highly non-trivial to collect, and should be considered a main contribution in itself.

**Questions**

As this is an emergency review, after the reviewer discussion period, I am not sure if the authors will be able to reply. But if they are able, I would just ask that they respond to the various points made in previous sub-sections.

**Limitations**

This paper did involve a fairly intense human subjects experiment for data collection, but the paper reports it was IRB-approved. Psychometrics of this kind are fairly common; I do not think the paper introduces any new potential negative societal impacts.

**Flag For Ethics Review**: No ethics review needed.

**Rating**: 4: Borderline reject: Technically solid paper where reasons to reject, e.g., limited evaluation, outweigh reasons to accept, e.g., good evaluation. Please use sparingly.

**Confidence**: 4: You are confident in your assessment, but not absolutely certain. It is unlikely, but not impossible, that you did not understand some parts of the submission or that you are unfamiliar with some pieces of related work.

---

> ### Comment · Program_Chairs · 2024-08-16
>
> **Weaknesses**
>
> The paper is vague about its technical contributions. It makes the following hedged novelty statement: "the first multi-task GP latent variables model _for dynamic idiographic assessment_". I took this to mean that the proposed approach is not very new technically, but it has never yet been applied to dynamic idiographic assessments. However, the paper also makes the general claim that it "advances the literatures on Gaussian process latent variable models...". I am of the opinion that tailoring existing modeling frameworks to new applications does provide a technical contribution, as it contributes a new view and set of interpretations/metaphors that can help to better understand the abstract model. I think the paper does contribute in this way. But I am unsure if the paper further contributes more substantially to the area of Gaussian process latent variable models, as the paper is vague about that.
>
> There is sloppiness in some of the main equations. For instance, equation 2 confuses a distribution with a random variable stating $p(\mathbf{f}^{(i}) \sim \cdots$, and I believe equation 3 confuses an inner with an outer product (shouldn't it be $w_i w_i^\top$?). There's also some confusing overloading of symbols between the background and the model sections. The background sets up the idea that an ideographic approach has distinct loadings $w_j^{(i)}$ different across $i$. But the proposed model itself does not seem to sport this as each $w_j$ is global for each task. I believe effectively the model does achieve an idiographic interpretation through the unit-specific kernel, but the connection between the background and model section is not clear.
>
> I think the second experiment has some major flaws. The paper claims this experiment validates the "Big Five" because model performance peaks at $K=5$. However, the model is only fit for $K=1...5$. We can clearly see from the log likelihood numbers that model performance for IPGP is monotonically increasing in $K$ for the values considered, suggesting that it will likely be even better at larger values of $K$; if that were true, it would not validate the "Big Five" theory. As this is an applied paper, I would expect a much higher level of rigor on one of the two applied case studies.

---

> ### Author Response · Authors · 2024-08-16
>
> Dear Reviewer 5m3j,
>
> Thanks for your valuable feedback. Please see our response below.
>
> > If I am reading the paper correctly, each subject was asked to complete a survey six times per day.
>
> Yes, that's correct. Collecting this data was a significant undertaking, and to our knowledge, we are among the very few groups that have ever collected data of such scope.
>
> > The paper is vague about its technical contributions. I am unsure if the paper further contributes more substantially to the area of Gaussian process latent variable models.
>
> We describe our model as a "latent variable GP model" since it belongs to a broader category of models involving latent variables and GPs (including but not exclusive to the famous GPLVM from Lawrence). Technically, our model also differs with GPLVM in that: (1) optimizes the factor loading matrix while marginalizing the latent variables, (2) accommodates categorical data through a non-Gaussian ordered logistic likelihood. Please see our other rebuttals for more clarification on our contributions.
>
> > Equation 2 confuses a distribution with a random variable stating $𝑝(𝑓^{(𝑖)})$, and I believe equation 3 confuses an inner with an outer product (shouldn’t it be $𝑤_𝑖𝑤_𝑖^⊤$?).
>
> Thank you for pointing out the typo in eq 2. We will fix it in revision. Regarding eq 3, please see our discussion of updated notations below.
>
> > The background sets up the idea that an ideographic approach has distinct loadings $𝑤_j^{(𝑖)}$ different across 𝑖. But the proposed model itself does not seem to sport this as each $𝑤_𝑗$ is global for each task.
>
> Thank you for bringing this up. We agree the notation could be improved -- the confusion here might come from overloading the use of variable w depending on its indexing. We will rename $w_i$ in Figure 1 and eq 3 to $Z^{(i)}$. The full updated notation is listed below:
>
> - Throughout our notation, superscript $(i)$ indicates unit and underscript $j$ indicates task.
> - $W_{\text{pop}}$ represents the K x J (K latent dimensions, J tasks) shared interpersonal loading matrix.
> - $Z^{(i)}$ represents the $K^*$ by J unit-specific low-rank loading matrix that serves to be the additional idiographic component, independent of $W_{\text{pop}}$. The rank $K^*$ of $Z^{(i)}$ need not be the same as the rank K of $W_{\text{pop}}$, and in fact we used $K^*=1$ in our experiments (see below).
> - $K_{\text{task}}^{(i)}$ is the unit-specific task covariance matrix with shared component $W_{\text{pop}}'$ $W_{\text{pop}}$ and unit-specific deviations ${Z^{(i)}}' Z^{(i)}$ of $K^*<K$ (with this revised notation the transpose in equation (3) is correct as written).
>
> Now $W_{\text{pop}}$ is a global parameter estimated for the entire population while $Z^{(i)}$s are unit-level parameters.  These are combined in eq (3) to create a unique kernel for each unit reflecting both components. To aid identification and performance, we focused on $K^*=1$ in the experiments; we also tried $K^*=2$ only to find degraded performance but extra computational costs.  Note that there is much less data to estimate unit-level deviations than there is to estimate population-level structure, so in general we might expect to take $K^* < K$ when modeling. However, the reviewer is correct that even in this rank-1 case, our setup still induces a unique $K_{\text{task}}^{(i)}$ as specified in eq 3. We will clarify the discussion here and include a list of notations in the appendix.
>
> > The paper claims this experiment validates the “Big Five” because model performance peaks at 𝐾=5. However, the model is only fit for 𝐾=1...5.
>
> Thank you for this valuable comment; we ran two additional factor analysis experiments accordingly. We first examine the performance of IPGP with higher model ranks $K \ge 5$ in the second case study (LOOPR); the results are shown in the table below.
>
> The results consistently support the model with rank precisely 5, which has both higher model evidence and lower BIC, providing much more convincing evidence for the Big Five theory than originally presented. In particular, the BIC has a decreasing trajectory from lower rank to 5 and an increasing one from 5 to higher rank, suggesting that increasing the rank to 5 is necessary for the model to capture the ideal structure of the data, but that exceeding this rank only overfits.
>
> **Performance of IPGP with ranks from 1 to 10 for LOOPR.**
> | RANK | 1 | 2 | 3 | 4 | 5 | 6 | 7 | 8 | 9 | 10 |
> | - | - | - | - | - | - | - | - | - | - | - |
> | **LL/N** ↑ | -1.478 | -1.477 | -1.477 | -1.477 | **-1.476** | -1.477 | -1.477 | -1.477 | -1.478 | -1.477 |
> | **BIC** (×10^11) ↓ | 1.2736 | 1.2726 | 1.2728 | 1.2726 | **1.2722** | 1.2725 | 1.2726 | 1.2732 | 1.2732 | 1.2734 |
>
> Please also see the related additional study described in our rebuttal to reviewer A4W7, where we varied the model rank among {2, 5, 8} in our simulation study (the true rank is 5). We saw similar results there, where the true rank consistently yielded the best model fit.

---

### Decision · Program_Chairs · 2024-09-25

**Decision:**

Accept (poster)

**Comment:**

NOTE: One reviewer provided vague and highly formulaic criticism that they were unable or unwilling to defend during the discussion period; I am ignoring their review in making my recommendation.

This is a very interesting applied paper, that connects recent methodology on multi-output Gaussian processes to a long-standing debate in psychometrics on the whether "features like personality share a common structure across the population, vary uniquely for individuals, or some combination"---i.e., the "nomothetic" versus "idiographic" approach.

The paper (and authors) should be credited for introducing a novel longitudinal data set composed of survey responses from N=93 subjects who were asked to fill out a personality assessment six times per day for 3 weeks. This data seems very labor-intensive to collect, and pairs well with the proposed model and theoretical question.

There was back-and-forth between the authors and reviewers about the proposed model, whether it was novel outside its applied domain, and whether it can be said to contribute broadly to the area of GPs and GP latent variable models. The paper hedges in several places about the novelty of the model, generally saying that it is novel for this (fairly narrow) applied domain, and the authors did not clarify this in their responses. The authors did however agree in the the discussion period to walk back some statements in the paper about whether the proposed model advances general methodology in GPLVMs. All-in-all, the discussion period clarified that the technical novelty of the model itself should not be overstated, but that the model nevertheless represented a non-trivial and correct adaptation of current methodology to an interesting applied problem, and was a contribution to the literature in the sense of providing a new illustration and use-case of multi-output GPs.

The authors cleared up several other problems raised by the reviewers during the discussion period. The reviewers detected some sloppiness in the equations; however, the discussion revealed that these were all minor typos which were easy to fix, and did not impede understanding of the technical content. One reviewer pointed out a major flaw in the main applied experiment which purported to validate the "Big Five" personality trait by running the model with K=1...5, and showing that K=5 performed best; the flaw was that the paper did not report values of K beyond 5. However, in their response, the authors provided supplemental experimental evidence that showed that the model's performance did in fact decline for K > 5.

I am in favor of acceptance. From the strict perspective of advancing general-purpose GP methodology, this paper is not very strong. However, the paper stands as a good example of translational applied work. It encodes an ongoing theoretical debate in psychology into a non-trivial multi-output GP model and applies the model to a novel data set that was difficult to obtain and bears directly on the applied question. I think NeurIPS should welcome work like this.

In the event of acceptance, the authors should make the following revisions to their camera-ready version:
- Any statements about the connection to GPLVMs should be made more precise and edited in accordance to the discussion with reviewers.
- Any novelty statements about the proposed model should be made more precise and/or omitted. If the model is not novel generally, but novel to this particular application, then this should be stated clearly, and citations to papers that employ the same (or very similar) models to other domains should be cited in the related work section.
- The typos in equations that reviewers pointed out should be fixed.
- The supplementary experiments reported during the discussion phase should be incorporated into the paper.